# Ordinal Causal Discovery

**Yang Ni**[1]                                    **Bani Mallick**[1]

[1]Department of Statistics, Texas A&M University, College Station, Texas, USA

## Abstract

Causal discovery for purely observational, categorical data is a long-standing challenging problem. Unlike continuous data, the vast majority of existing methods for categorical data focus on inferring the Markov equivalence class only, which leaves the direction of some causal relationships undetermined. This paper proposes an identifiable ordinal causal discovery method that exploits the ordinal information contained in many real-world applications to uniquely identify the causal structure. The proposed method is applicable beyond ordinal data via data discretization. Through real-world and synthetic experiments, we demonstrate that the proposed ordinal causal discovery method combined with simple score-and-search algorithms has favorable and robust performance compared to state-of-the-art alternative methods in both ordinal categorical and non-categorical data. An accompanied R package `OCD` is freely available at the first author's website.

## 1 INTRODUCTION

Causal discovery [Spirtes et al., 2000, Pearl, 2009] is becoming increasingly more popular in machine learning and finds numerous applications, e.g., biology [Sachs et al., 2005], psychology [Steyvers et al., 2003], and neuroscience [Shen et al., 2020], of which the prevailing goal is to discover causal relationships of variables of interest. The discovered causal relationships are useful for predicting a system's response to external interventions [Pearl, 2009], a key step towards understanding and engineering that system. While the gold standard for causal discovery remains the controlled experimentation, it can be too expensive, unethical, or even impossible in many cases, particularly on human beings. Therefore, inferring the unknown causal structures of complex systems from purely observational data is often desirable and, sometimes, the only option.

This paper considers causal discovery for ordinal categorical data. Categorical data are common across multiple disciplines. For example, psychologists often use questionnaires to measure latent traits such as personality and depression. The responses to those questionnaires are often categorical, say, with five levels (5-point Likert scale): "strongly disagree", "disagree", "neutral", "agree", and "strongly agree". In genetics, single-nucleotide polymorphisms are categorical variables with three levels (mutation on neither, one, or both alleles). Categorical data also arise as a result of discretization of non-categorical (e.g., continuous and count) data. For instance, in biology, gene expression data are often trichotomized to "underexpression", "normal expression", and "overexpression" [Parmigiani et al., 2002, Pe'er, 2005, Sachs et al., 2005] in order to reduce sequencing technical noise while retaining biological interpretability.

While causal discovery for purely observational categorical data have been extensively studied, the vast majority of existing methods [Heckerman et al., 1995, Chickering, 2002] have exclusively focused on Bayesian networks (BNs) with nominal (unordered) categorical variables. It has been well established that a nominal/multinomial BN is generally only identifiable up to *Markov equivalence class* in which all BNs encode the same Markov properties. For example, $X \rightarrow Y$ and $Y \rightarrow X$ are Markov equivalent and also distribution equivalent [Spirtes and Zhang, 2016] with a multinomial likelihood; therefore, they are non-identifiable with purely observational data.

In many real-world applications, categorical data (including the aforementioned Likert scale, single-nucleotide polymorphisms, and discretized gene expression data) contain ordinal information. In this paper, we show that this often-overlooked ordinal information is crucial in causal discovery for categorical data. We propose an ordinal causal discovery (OCD) method via an ordinal BN. Assuming causal Markov and causal sufficiency, we prove OCD to be identifiable

*Accepted for the 38th Conference on Uncertainty in Artificial Intelligence* (UAI 2022).

in general for ordinal categorical data. Score-and-search BN structure learning algorithms are developed – exhaustive search for small networks (e.g., bivariate data) and greedy search for moderate-sized networks. Through extensive experiments with real-world and synthetic datasets, we demonstrate that the proposed OCD is identifiable, robust, applicable to both categorical and non-categorical data, and competitive against a range of state-of-the-art causal discovery methods. To the best of our knowledge, we are the first to exploit the ordinal information for causal discovery in categorical data. Our major contributions are four-fold.

1. We advocate the usefulness of ordinal information of categorical data in causal discovery, which has been overlooked in the literature.

2. We propose the first causal discovery method, OCD, for ordinal categorical data.

3. We prove that OCD is generally identifiable for bivariate data, in contrast to the non-identifiability of multinomial BNs.

4. We demonstrate the strong utility of OCD by comparison with state-of-the-art alternatives using real-world and synthetic datasets.

## 1.1 RELATED WORK

For brevity, we review causal discovery methods that are fully identifiable with observational data.

**Non-Categorical Data.** Model-based BNs for continuous data are often represented as additive noise models. Under such representation, BNs are generally identifiable if the noises are non-Gaussian [Shimizu et al., 2006], if the functional form of the additive noise model is nonlinear [Hoyer et al., 2009, Zhang and Hyvärinen, 2009], or if the noise variances are equal [Peters and Bühlmann, 2014]. Also see much of the recent literature that focuses on bivariate causal discovery [Mooij et al., 2010, Janzing et al., 2012, Chen et al., 2014, Sgouritsa et al., 2015, Hernandez-Lobato et al., 2016, Marx and Vreeken, 2017, Blöbaum et al., 2018, Marx and Vreeken, 2019, Tagasovska et al., 2020]. For count data, Park and Raskutti 2015 proposed a Poisson BN and showed that it is identifiable based on the overdispersion property of Poisson BNs. By replacing overdispersion property with constant moments ratio property, Park and Park 2019 extended Poisson BNs to the generalized hypergeometric family which contains many count distributions such as binomial, Poisson, and negative binomial. Recently, Choi et al. 2020 developed a zero-inflated Poisson BN for zero-inflated count data.

**Categorical Data.** For nominal categorical data, causal identification is possible under certain assumptions [Peters et al., 2010, Suzuki et al., 2014, Liu and Chan, 2016, Cai et al., 2018, Compton et al., 2020, Qiao et al., 2021], e.g.,

when the categories admit hidden compact representations or when data follow a discrete additive noise model. However, to the best of our knowledge, causal discovery for ordinal data, which are very common in practice, has not been studied. Whether a categorical variable is ordinal or not is, in our opinion, easier to comprehend than the aforementioned assumptions of categorical data (e.g., discrete additive noise). We remark that a recent paper [Luo et al., 2021] also considered ordinal data. However, their work is substantially different from ours. The most prominent difference is that the causal graph of Luo et al. [2021] is only identifiable up to Markov equivalence classes whereas the proposed method is uniquely identifiable, which is proved for the bivariate case.

**Mixed Data.** There are recent developments for mixed data causal discovery [Cui et al., 2018, Tsagris et al., 2018, Sedgewick et al., 2019], some of which include categorical data. However, the ordinal nature of the categorical data is not exploited for causal identification; therefore, these algorithms output Markov equivalent BNs instead of individual BNs. The latent variable approach by Wei et al. [2018] could in principle be extended to ordinal data. However, the causal Markov assumption of latent variables cannot translate to the observed variables and the inferred causality does not have direct causal interpretation on the observed variables.

## 2 BIVARIATE ORDINAL CAUSAL DISCOVERY

We first introduce the proposed OCD method for bivariate data, which will be extended to multivariate data in Section 4. Let $(X, Y) \in \{1, \ldots, S\} \times \{1, \ldots, L\}$ denote a pair of ordinal variables with $S$ and $L$ levels, of which the possible causal relationships, $X \to Y$ or $Y \to X$, are under investigation. Throughout the paper, we make the causal Markov and causal sufficient assumptions, which are frequently adopted in the causal discovery literature [Pearl, 2009]. The former allows us to interpret the proposed model causally (beyond conditional independence) whereas the latter asserts that there are no unmeasured confounders.

The bivariate OCD considers the following probability distribution for causal model $X \to Y$,

$$p_{X \to Y}(X, Y) = p(X)p(Y|X), \quad (1)$$

where $p(X)$ is a multinomial/categorical distribution with probabilities $\boldsymbol{\pi} = (\pi_1, \ldots, \pi_S)$ with $\sum_{s=1}^{S} \pi_s = 1$, and $p(Y|X)$ is defined by an ordinal regression model [Agresti, 2003],

$$Pr(Y \le \ell | X) = F(\gamma_\ell - \beta_X), \quad \ell = 1, \ldots, L, \quad (2)$$

where $\beta_X$ is a generic notation of $\beta_1, \ldots, \beta_S$ for $X = 1, \ldots, S$. Typical choices of the link function $F$ are probit and inverse logit, which are empirically quite similar;

hereafter we always use the probit link except for the identifiability theory, which is valid for both link functions. We fix $\gamma_1 = 0$ for ordinal regression parameter identifiability [Agresti, 2003]. Equation (2) implies the conditional probability distribution $Pr(Y = \ell | X = s) = F(\gamma_\ell - \beta_s) - F(\gamma_{\ell-1} - \beta_s)$ for $\ell = 1, \ldots, L$ and $s = 1, \ldots, S$ where $\gamma_0 = -\infty$ and $\gamma_L = \infty$. Let $\boldsymbol{\beta} = (\beta_1, \ldots, \beta_S)$ and $\boldsymbol{\gamma} = (\gamma_2, \ldots, \gamma_{L-1})$. We denote the model $p_{X \to Y}$ by $p_{X \to Y}(X, Y | \boldsymbol{\pi}, \boldsymbol{\beta}, \boldsymbol{\gamma})$. Similarly, we define the probability model $p_{Y \to X}$ as $p_{Y \to X}(Y, X | \boldsymbol{\rho}, \boldsymbol{\alpha}, \boldsymbol{\eta})$. If the maximum likelihood estimate $\widehat{p}_{X \to Y}$ given observations of $(X, Y)$ is strictly larger than $\widehat{p}_{Y \to X}$, then $X \to Y$ is deemed a more likely data generating causal model.

# 3 IDENTIFIABILITY

We will show that the proposed OCD is generally identifiable.

**Definition 1 (Distribution Equivalence)**
$p_{X \to Y}(X, Y | \boldsymbol{\pi}, \boldsymbol{\beta}, \boldsymbol{\gamma})$ and $p_{Y \to X}(Y, X | \boldsymbol{\rho}, \boldsymbol{\alpha}, \boldsymbol{\eta})$ *are distribution equivalent if for any values of $(\boldsymbol{\pi}, \boldsymbol{\beta}, \boldsymbol{\gamma})$ there exist values of $(\boldsymbol{\rho}, \boldsymbol{\alpha}, \boldsymbol{\eta})$ such that $p_{X \to Y}(X, Y | \boldsymbol{\pi}, \boldsymbol{\beta}, \boldsymbol{\gamma}) = p_{Y \to X}(Y, X | \boldsymbol{\rho}, \boldsymbol{\alpha}, \boldsymbol{\eta})$ for any $X, Y$, and vice versa.*

Distribution equivalent causal models are clearly not distinguishable from each other by examining their observational distributions. The well-known multinomial BNs are distribution equivalent as illustrated in the following example.

**Example 1 (Multinomial BN)** *Consider a bivariate multinomial BN of $X \to Y$ whose conditional $p(Y|X)$ and marginal $p(X)$ probability distributions are given in Figure 1(a), and the joint distribution $p(X, Y)$ is given in Figure 1(b). Because of the multinomial assumption, we can find a set of parameters, i.e., the conditional $p(X|Y)$ and marginal $p(Y)$ probabilities (Figure 1(c)) of the reverse causal model $Y \to X$, which leads to the same joint distribution. Therefore, the probability distribution does not provide information for causal identification.*

Incorporating the underappreciated ordinal information, we will show that $p_{X \to Y}(X, Y | \boldsymbol{\pi}, \boldsymbol{\beta}, \boldsymbol{\gamma})$ and $p_{Y \to X}(Y, X | \boldsymbol{\rho}, \boldsymbol{\alpha}, \boldsymbol{\eta})$ are generally *not* distribution equivalent and are, therefore, identifiable.

**Theorem 1 (Identifiability of OCD)** *Let $X \in \{1, \ldots, S\}$ and $Y \in \{1, \ldots, L\}$ where $S, L > 2$. Suppose $X \to Y$ is the data generating causal model and the observational probability distribution of $(X, Y)$ is given by*

$$p(X, Y) = p_{X \to Y}(X, Y | \boldsymbol{\pi}, \boldsymbol{\beta}, \boldsymbol{\gamma}).$$

*For almost all $(\boldsymbol{\pi}, \boldsymbol{\beta}, \boldsymbol{\gamma})$ with respect to the Lebesgue measure, the distribution cannot be equivalently represented by*

*the reverse causal model, i.e., there does not exist $(\boldsymbol{\rho}, \boldsymbol{\alpha}, \boldsymbol{\eta})$ such that,*

$$p(X, Y) = p_{Y \to X}(Y, X | \boldsymbol{\rho}, \boldsymbol{\alpha}, \boldsymbol{\eta}), \forall X, Y.$$

The proof based on properties of real analytic functions is provided in the Supplementary Materials. We demonstrate Theorem 1 by revisiting Example 1.

**Example 2 (Ordinal BN)** *The conditional $p(Y|X)$ and marginal $p(X)$ probability distributions in Figure 1(a) coincide with those under the ordinal BN $p_{X \to Y}(X, Y | \boldsymbol{\pi}, \boldsymbol{\beta}, \boldsymbol{\gamma})$ with $\boldsymbol{\pi} = (0.25, 0.25, 0.5)$, $\gamma = 1$, and $\boldsymbol{\beta} = (1, -1, 1)$. Given a large enough dataset, the MLE of $p(X, Y)$ can be arbitrarily close to that in Figure 1(b). However, there does not exist any set of parameter values in the reverse causal model $p_{Y \to X}(Y, X | \boldsymbol{\rho}, \boldsymbol{\alpha}, \boldsymbol{\eta})$ that produces the conditional $p(X|Y)$ and marginal $p(Y)$ probability distributions in Figure 1(c). Therefore, the reverse causal model $p_{Y \to X}(Y, X | \boldsymbol{\rho}, \boldsymbol{\alpha}, \boldsymbol{\eta})$ cannot adequately fit the data generated from $p_{X \to Y}(X, Y | \boldsymbol{\pi}, \boldsymbol{\beta}, \boldsymbol{\gamma})$. For example, even with 100,000 observations, the MLE of $p(X, Y)$ under $p_{Y \to X}(Y, X | \boldsymbol{\rho}, \boldsymbol{\alpha}, \boldsymbol{\eta})$ still has a large bias (Figure 1(d)), which will never approach 0. Therefore, $p_{X \to Y}(X, Y | \boldsymbol{\pi}, \boldsymbol{\beta}, \boldsymbol{\gamma})$ can be distinguished from $p_{Y \to X}(Y, X | \boldsymbol{\rho}, \boldsymbol{\alpha}, \boldsymbol{\eta})$.*

Note that Theorem 1 excludes the binary variable case, under which OCD is not identifiable. This is expected because there is no difference between ordinal and nominal categorical variables in this case; the latter is known to be non-identifiable.

# 4 EXTENSION TO MULTIVARIATE ORDINAL CAUSAL DISCOVERY

While the vast majority of the existing identifiable causal discovery methods for categorical data [Peters et al., 2010, Suzuki et al., 2014, Liu and Chan, 2016, Cai et al., 2018, Compton et al., 2020] have primarily focused on bivariate cases, we extend the proposed bivariate OCD to multivariate data. Let $\boldsymbol{X} = (X_1, \ldots, X_q) \in \{1, \ldots, L_1\} \times \cdots \times \{1, \ldots, L_q\}$ denote $q$ ordinal variables. Let $G = (V, E)$ denote a causal BN with a set of nodes $V = \{1, \ldots, q\}$ representing $\boldsymbol{X}$ and directed edges $E \subset V \times V$ representing direct causal relationships (with respect to $\boldsymbol{X}$). Let $pa(j) = \{k | k \to j\} \subseteq V$ denote the set of direct causes (*parents*) of node $j$ in $G$ and let $\boldsymbol{X}_{pa(j)} = \{X_k | k \in pa(j)\}$. Given $G$, the joint distribution of $\boldsymbol{X}$ factorizes,

$$p(\boldsymbol{X} | G) = \prod_{j=1}^{q} p\left(X_j | \boldsymbol{X}_{pa(j)}\right), \tag{3}$$

where each conditional distribution $p\left(X_j | \boldsymbol{X}_{pa(j)}\right)$ is an ordinal regression model of which the cumulative distribution

| $P(Y\mid X)$ | $X=1$ | $X=2$ | $X=3$ |
|---|---|---|---|
| $Y=1$ | 0.16 | 0.84 | 0.16 |
| $Y=2$ | 0.34 | 0.14 | 0.34 |
| $Y=3$ | 0.50 | 0.02 | 0.50 |

$\times$

| $X$ | $P(X)$ |
|---|---|
| 1 | 0.25 |
| 2 | 0.25 |
| 3 | 0.50 |

| $P(X,Y)$ | $X=1$ | $X=2$ | $X=3$ |
|---|---|---|---|
| $Y=1$ | 0.040 | 0.210 | 0.080 |
| $Y=2$ | 0.085 | 0.035 | 0.170 |
| $Y=3$ | 0.125 | 0.005 | 0.250 |

(a)     (b)

| $P(X\mid Y)$ | $Y=1$ | $Y=2$ | $Y=3$ |
|---|---|---|---|
| $X=1$ | 0.12 | 0.29 | 0.33 |
| $X=2$ | 0.64 | 0.12 | 0.01 |
| $X=3$ | 0.24 | 0.59 | 0.66 |

$\times$

| $Y$ | $P(Y)$ |
|---|---|
| 1 | 0.33 |
| 2 | 0.29 |
| 3 | 0.38 |

| $\widehat{P}(X,Y)$ | $X=1$ | $X=2$ | $X=3$ |
|---|---|---|---|
| $Y=1$ | 0.104 | 0.087 | 0.141 |
| $Y=2$ | 0.065 | 0.071 | 0.154 |
| $Y=3$ | 0.078 | 0.089 | 0.211 |

(c)     (d)

Figure 1: Illustration. (a) Conditional $p(Y|X)$ and marginal $p(X)$ probability distributions. They coincide with those under $p_{X\to Y}(X,Y|\boldsymbol{\pi},\boldsymbol{\beta},\boldsymbol{\gamma})$ with $\boldsymbol{\pi}=(0.25,0.25,0.5)$, $\gamma=1$, and $\boldsymbol{\beta}=(1,-1,1)$. (b) The joint distribution $p(X,Y)=p(X)p(Y|X)$. (c) Conditional $p(X|Y)$ and marginal $p(Y)$ probability distributions from the same joint distribution $p(X,Y)$. (d) Maximum likelihood estimate of $p(X,Y)$ under $p_{Y\to X}(Y,X|\boldsymbol{\rho},\boldsymbol{\alpha},\boldsymbol{\eta})$ using data generated from $p(X,Y)$ in (b) with sample size 100,000.

is given by, for $\ell=1,\ldots,L_j$,

$$Pr(X_j \leq \ell|\boldsymbol{X}_{pa(j)}) = F\left(\gamma_{j\ell} - \sum_{k\in pa(j)}\beta_{jkX_k} - \alpha_j\right),$$

where $\alpha_j$ is the intercept and $\beta_{jkX_k}$ is a generic notation of $\beta_{jk1},\ldots,\beta_{jkL_k}$ for $X_k=1,\ldots,L_k$. We set $\gamma_{j1}=\beta_{jkL_k}=0$ for ordinal regression parameter identifiability [Agresti, 2003]. The implied conditional probability distribution is given by,

$$Pr(X_j = \ell|\boldsymbol{X}_{pa(j)}=\boldsymbol{s}) = F(\gamma_{j\ell} - \sum_{k\in pa(j)}\beta_{jkh_k} - \alpha_j)$$
$$- F(\gamma_{j,\ell-1} - \sum_{k\in pa(j)}\beta_{jkh_k} - \alpha_j),$$

for $\ell=1,\ldots,L_j$ and $\boldsymbol{s}\in\prod_{k\in pa(j)}\{1,\ldots,L_k\}$. In summary, the multivariate OCD model is parameterized by $\boldsymbol{\gamma}_j=(\gamma_{j2},\ldots,\gamma_{j,L_j-1})$, $\boldsymbol{\beta}_{jk}=(\beta_{jk1},\ldots,\beta_{jk,L_k-1})$, and $\alpha_j$, for $j=1,\ldots,q$ and $k\in pa(j)$.

## 5 CAUSAL GRAPH STRUCTURE LEARNING

We develop simple score-and-search learning algorithms to estimate the structure of causal graphs, which already show strong empirical performance (see Section 6), although more sophisticated learning methods such as Bayesian inference could be adopted to further improve the performance.

**Score.** We score causal graphs by the Bayesian information criterion (BIC). We choose BIC over AIC because it favors a more parsimonious causal graph due to the heavier penalty on model complexity and generally has a better empirical performance. Let $\boldsymbol{x}=(\boldsymbol{x}_1,\ldots,\boldsymbol{x}_n)$ denote $n$ realizations of $\boldsymbol{X}$. The score of $G$ (smaller is better) is given by

$$\mathrm{BIC}(G|\boldsymbol{x}) = -2\sum_{i=1}^{n}\log\widehat{p}(\boldsymbol{x}_i|G) + K\log(n),$$

where $K$ is the number of model parameters and $\widehat{p}(\boldsymbol{x}_i|G)$ is the joint distribution (3) evaluated at $\boldsymbol{x}_i$ given the MLE of model parameters.

**Exhaustive Search.** For small networks (say $q=2$ or 3), we compute the scores for all networks $\mathcal{G}$, and identify $\widehat{G}=\arg\min_{G\in\mathcal{G}}\mathrm{BIC}(G|\boldsymbol{x})$. While this approach is exact and useful for bivariate OCD, it becomes computationally infeasible for moderate-sized networks as the number of networks $|\mathcal{G}|$ grows super-exponentially in $q$.

**Greedy Search.** We use a simple iterative greedy search algorithm [Chickering, 2002, Scutari et al., 2019] for moderate-sized networks. At each iteration, we score all the graphs that can be reached from the current graph by an edge addition, removal, or reversal. We replace the current graph by the graph with the largest improvement (largest decrease in BIC) and stop the algorithm when the score can no longer be improved. The greedy search algorithm is summarized in Algorithm 1, which is guaranteed to find a local optimal graph. The algorithm can be improved by tabu search and random non-local moves [Scutari et al., 2019] but we do not pursue this direction as the simple greedy algorithm already yields favorable results against state-of-the-art alternative methods. The worst per iteration cost is $O(qf(n,m,L))$ for $q$ nodes, $n$ observations, $m$ maximum number of parents, and $L=\max_j L_j$ maximum levels, where $f(n,m,L)$ is the computational complexity of an ordinal regression with $m$ regressors. This is because at most $2q$ score evaluations are required at each iteration [Scutari

**Algorithm 1** Greedy Search

**Input:** data $\boldsymbol{x}$, initial empty graph $G$
Compute BIC$(G|\boldsymbol{x})$ and set BIC$_\star$=BIC$(G|\boldsymbol{x})$.
**repeat**
    Initialize $Improvement = false$.
    **for** all graphs $G'$ reachable from $G$ **do**
        Compute BIC$(G'|\boldsymbol{x})$.
        **if** BIC$(G'|\boldsymbol{x})<$ BIC$_\star$ **then**
            Set $G=G'$ and BIC$_\star$=BIC$(G'|\boldsymbol{x})$
            Set $Improvement = true$.
        **end if**
    **end for**
**until** $Improvement$ is $false$
**Output:** graph $G$

---

et al., 2019]. We use `polr` function in the R package `MASS` for ordinal regression which appears to scale linearly in $n, m$, and $L$, empirically.

# 6 EXPERIMENTS

We evaluate the proposed and state-of-the-art alternative causal discovery methods with synthetic as well as three sets of real data. The real data are not categorical and therefore allow us to extend our comparison to causal models designed for continuous data.

## 6.1 SYNTHETIC ORDINAL DATA

We simulate low-dimensional, higher-dimensional, and bivariate (with confounders) synthetic ordinal data.

### 6.1.1 Low-Dimensional Multivariate Ordinal Data

We consider synthetic ordinal data ($n = 500, q = 10$). To mimic survey data with 5-point Likert-scale questionnaires, we simulate data from the proposed OCD model with $L_j = L = 5, \forall j$. The true BN is generated randomly (Figure 2(a)), which has one v-structure (i.e., subgraph $j \to k \leftarrow i$). Its Markov equivalence class, represented by a completed partially directed acyclic graph (CPDAG), can be obtained by removing the directionality of the red dashed edges in Figure 2(a). We consider 6 scenarios with different levels of signal strength by generating simulation true $\beta_{jk\ell}$'s and $\alpha_j$'s independently from $N(0, \sigma^2)$ with $\sigma = 0.25, 0.5, 0.75, 1, 1.25, 1.5$. Parameters $\gamma_{j\ell}$'s are chosen to have balanced class size for each variable.

**Implementations.** Standard causal discovery methods for categorical data are multinomial BNs with BIC or BDe score, which discard the ordinal information and therefore only estimate the Markov equivalence classes. They are implemented using model averaging with 500 bootstrapped

samples (page 145, Scutari and Denis 2014). We compare them with the proposed OCD, all implemented using greedy search. In addition, we also consider a two-step procedure [Friedman and Koller, 2003] and a recent ordinal structural equation model [Luo et al., 2021, OSEM]. The two-step procedure first learns a causal ordering and then estimates the causal multinomial BN given the ordering based on BIC (called "BIC+" hereafter). This procedure outputs an estimated BN. The OSEM introduces latent Gaussian variables, on which a structural equation model is imposed. Like multinomial BNs with BIC or BDe score, OSEM identifies the Markov equivalence classes. The tuning parameter of OSEM is set to 1.

**Metrics.** We compute the structural hamming distance (SHD) and the structural intervention distance (SID) with R package `SID`. The SHD between two graphs is the number of edge additions, deletions, or reversals required to transform one graph to the other. The SID measures "closeness" between two causal graphs in terms of their implied intervention distributions (see Peters and Bühlmann 2015 for the formal definition). Note that since multinomial BNs with BIC and BDe, and OSEM can only identify CPDAG, the smallest SHD that they can achieve is 5 (the number of undirected edges in the true CPDAG).

**Results.** The SHD and SID averaged over 5 repeat simulations are shown in Figure 2(b)-(c) as functions of signal strength $\sigma$. Since multinomial BNs with BDe and BIC, and OSEM only estimate CPDAGs, we report the lower bounds of their SID. There are several conclusions that can be drawn. First, OCD is empirically identifiable because both SHD and SID quickly approach 0 as signal becomes stronger. Second, OCD uniformly outperforms the alternative methods in both SHD and SID across all signal levels, which suggests that exploiting the ordinal nature of ordinal categorical data is crucial for causal discovery. Third, BIC+ is better than BIC and BDe in SHD but not necessarily in SID, suggesting the estimated causal ordering from BIC+ is biased. Fourth, although OSEM also accounts for ordinal data, it is not identifiable and may be sensitive to the tuning parameter, which is hard to be objectively tuned. Therefore, we drop OSEM in the subsequent simulations.

**Different Number of Categories.** In the Supplementary Materials, we present additional simulation scenarios with a different number $L = 3$ of categories. Similarly to the scenarios with $L = 5$, OCD significantly outperforms the competing methods.

### 6.1.2 Higher-Dimensional Multivariate Ordinal Data

We fix the sample size $n = 500$ and the number of categories $L = 5$ but vary the number of nodes $q = 10, 20, \ldots, 100$ and the signal strength $\sigma = 0.25, 0.5, 0.75, 1$. The graphs are kept at the same sparsity as in Section 6.1.1 across $q$

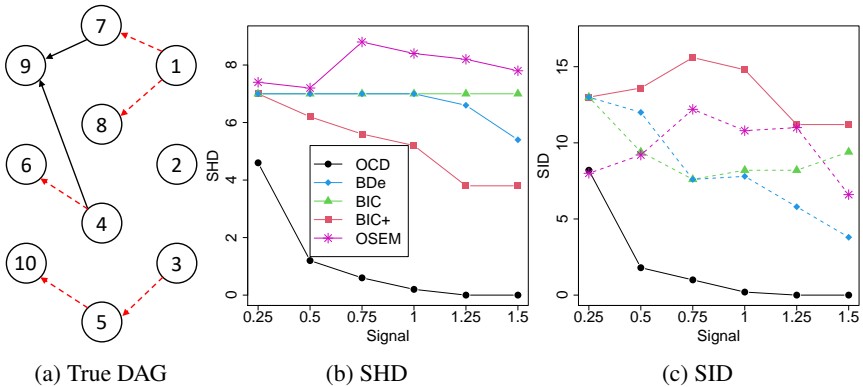

| | |
|---|---|
| (a) True DAG | (b) SHD (c) SID |

Figure 2: Synthetic ordinal data. The dashed lines in (c) are the lower bounds of SID of BDe, BIC, and OSEM, which output CPDAGs instead of BNs. Lower SHD and SID are better.

(denser graphs will be considered later). The SHD is shown in Figure 3 whereas the SID is provided in the Supplementary Materials. The proposed OCD uniformly outperforms the competing methods BDe, BIC, and BIC+ across $q$ and $\sigma$. In general, OCD is quite stable as $q$ increases when the signal strength is moderate to moderately large $\sigma \geq 0.5$ whereas the competing methods quickly deteriorate with $q$ regardless of the signal strength.

**Scalability.** We investigate the scalability of the proposed OCD with respect to $n, L$, and $q$. We vary $n = 500, 750, \cdots, 2750$ (keeping $q = 10$ and $L = 5$), $L = 5, \ldots, 14$ (keeping $n = 500$ and $q = 10$), and $q = 10, 20, \ldots, 100$ (keeping $n = 500$ and $L = 5$). The total CPU times in seconds on a 2.9 GHz 6-Core Intel Core i9 laptop are provided in the Supplementary Materials. The greedy search appears to scale linearly in $n$ and $L$, and quadratically in $q$, which agrees with the complexity analysis in Section 5. It is moderately scalable: e.g., for $q = 100$, the search completes in about 3 hours.

**Denser Graphs.** In the Supplementary Materials, we present additional simulation scenarios with denser graphs for $q = 50$ nodes and more v-structures, which lead to similar conclusions, i.e., OCD significantly outperforms the competing methods in SHD and SID.

### 6.1.3 Bivariate Ordinal Data with Unmeasured Confounders

While our identifiability theory assumes no unmeasured confounders, we now empirically test the sensitivity of OCD to unmeasured confounders for bivariate ordinal data. We generate trivariate ordinal data $(X_1, X_2, X_3)$ with $L = 5$ from the following true causal graph,

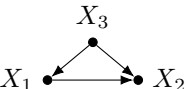

We hide $X_3$ as a confounder and apply OCD to $(X_1, X_2)$. In the simulation truth, we assume $\beta_{jk\ell}$, for each $\ell = 1, \ldots, L$, to be the same for all $j \neq k$, i.e., the confounding effect is the same as the causal effect, which is simulated from $N(0, \sigma^2)$. We consider different levels of signal strength $\sigma = 0.25, 0.5, 0.75, 1, 1.25, 1.5$ and different sample sizes $n = 100, 200, \ldots, 1000$. Under each combination of $(\sigma, n)$, we repeat the experiment 100 times, and report the average accuracy (ACC) for *forced decisions*. The forced decision forces methods to choose between $X_1 \to X_2$ and $X_2 \to X_1$. The same metric has been used in similar bivariate causal discovery problems [Mooij et al., 2016, Tagasovska et al., 2020]. OCD is relatively robust to confounders (Figure 4(a)): it is able to correctly identify the causal direction given a large enough sample size or when the signal is sufficiently strong. For comparison, we apply a recent causal discovery method for bivariate nominal categorical data, HCR [Cai et al., 2018]. Its average ACC is shown in Figure 4(b). We find the ACC of HCR is uniformly lower than that of OCD although we note that HCR is not specifically designed for this task.

### 6.2 SACHS'S SINGLE-CELL FLOW CYTOMETRY DATA

We evaluate the proposed OCD on the well-known single-cell flow cytometry dataset [Sachs et al., 2005], which contains measurements of $q = 11$ phosphorylated proteins under different experimental conditions. Sachs et al. 2005 provided a consensus causal network of these proteins, which could be used to gauge the performance of causal discovery algorithms. As in Tagasovska et al. 2020, we consider the *cd3cd28* dataset with 853 cells subject to the same experimental condition.

**Implementations.** Since the raw measurements are highly skewed and heavy-tailed, Sachs et al. 2005 discretized the data into $L = 3$ levels ("low", "average", and "high") and fit a multinomial BN based on the BDe score. As we will

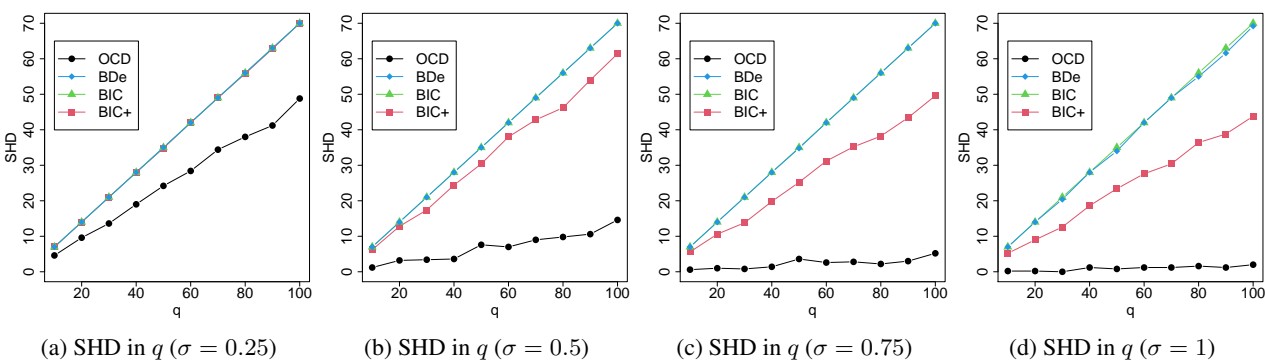

| (a) SHD in $q$ ($\sigma = 0.25$) | (b) SHD in $q$ ($\sigma = 0.5$) | (c) SHD in $q$ ($\sigma = 0.75$) | (d) SHD in $q$ ($\sigma = 1$) |

Figure 3: SHD (lower is better) for OCD, BDe, BIC, and BIC+ as functions of $q$ in the synthetic ordinal data with the sample size fixed at $n = 500$ and different signal strength $\sigma \in \{0.25, 0.5, 0.75, 1\}$.

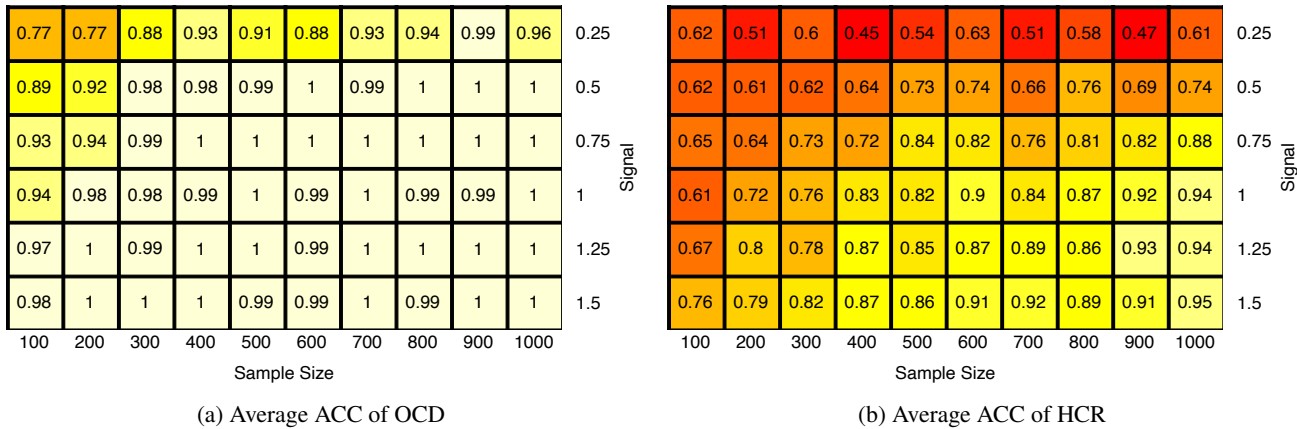

(a) Average ACC of OCD

(b) Average ACC of HCR

Figure 4: Synthetic ordinal data with confounders. Average ACC (higher is better) of (a) OCD and (b) HCR under different sample sizes and levels of signal strength.

see, this approach throws away the ordinal information inherent in the raw measurements and hence significantly underperforms OCD (with greedy search). For comparison, we also apply ANM [Hoyer et al., 2009], LiNGAM [Shimizu et al., 2006], RESIT with the Gaussian process implementation [Peters et al., 2014], bivariate causal discovery methods (HCR, bQCD [Tagasovska et al., 2020], GR-AN [Hernandez-Lobato et al., 2016], IGCI with uniform measure [Janzing et al., 2012], SLOPE [Marx and Vreeken, 2017]), and methods inferring Markov equivalence classes (PC [Spirtes et al., 2000], CPC [Ramsey et al., 2012], GES [Chickering, 2002], IAMB [Tsamardinos et al., 2003], multinomial BNs with BIC and BDe), and the mixed data approach MXM [Tsagris et al., 2018] to the raw continuous data. For bivariate causal discovery methods, we follow a similar *ad hoc* procedure in Tagasovska et al. 2020: first run CAM [Bühlmann et al., 2014] and then orient the estimated edges by the bivariate methods. HCR is the closest competitor as it is also designed for categorical data although with a very different scope (only applicable to bivariate nominal categorical data and assuming the existence of hidden compact representations). We still compare the proposed

OCD with OSEM. To address the tuning parameter issue of OSEM, we tune it in an oracle way on an evenly-spaced 12-grid from 0.5 to 6.0.

**Metrics.** We use the same SHD and SID metrics as in Section 6.1. For methods that output CPDAGs instead of BNs, we report the lower and upper bounds of SID.

**Results.** In Table 1, we summarize the SHD and SID. OCD shows very strong performance comparing to state-of-the-art alternatives. It has the lowest SHD and the second lowest SID, which shows benefit of discretization for highly noisy data. The substantial improvement of OCD from multinomial BN with BDe (SHD 14 vs 21) highlights the importance of exploiting the ordinal information of discrete data for causal discovery. While there is strong motivation (e.g., biological interpretation) to use $L = 3$ for this dataset, we test OCD with $L$ up to 10. OCD stays very competitive within this range: the SID remains 62 for all $L$ whereas the SHD slightly increases as $L$ increases possibly due to relatively small sample size, e.g., SHD $= 16$ for $L = 10$, which is still quite competitive (second to SHD $= 15$ for bQCD and IGCI). The smallest SHD that OSEM achieves

over the range of tuning parameter is 18.

Table 1: Sachs's data. Methods (marked by *) that are only applicable to bivariate data are combined with CAM. PC, CPC, GES, IAMB, BIC, BDe, and MXM only learn CPDAGs; we provide the lower and upper bounds of SID. Lower SHD and SID are better.

|  | OCD | bQCD* | IGCI* | GR-AN* |
|---|---|---|---|---|
| SHD | 14 | 15 | 15 | 16 |
| SID | 62 | 69 | 82 | 80 |
|  | HCR* | SLOPE* | ANM | LiNGAM |
| SHD | 16 | 17 | 17 | 17 |
| SID | 76 | 86 | 78 | 86 |
|  | PC | CPC | GES | IAMB |
| SHD | 18 | 18 | 18 | 20 |
| SID | 50-83 | 50-80 | 50-80 | 79-70 |
|  | BIC | BDe | MXM | RESIT |
| SHD | 20 | 21 | 21 | 40 |
| SID | 53-77 | 49-104 | 49-104 | 45 |

## 6.3 CAUSEEFFECTPAIRS (CEP) BENCHMARK DATA

We consider the CauseEffectPairs (CEP) benchmark data [Mooij et al., 2016] (version: 12/20/2017), which contain 108 datasets from 37 domains (e.g., biology, economy, engineering, and meteorology). Each dataset contains a pair of variables $(X, Y)$ for which the causal relationship is clear from the context, e.g., older "age" causes higher "glucose". We retain the same 99 pairs as in Tagasovska et al. 2020 that have univariate non-binary cause and effect variables.

**Implementations.** We compare OCD with HCR, bQCD, IGCI, CAM, SLOPE, LiNGAM, and RESIT. To apply OCD and HCR, we discretize each variable at $L - 1$ quantiles for $L \in \{10, \ldots, 20\}$. All other methods are applied to the (standardized) continuous data without discretization.

**Metrics.** We compute the ACC for forced decisions as in Section 6.1.3 and, additionally, the *area under the receiver operating curve* (AUC) for *ranked decision*. The ranked decision ranks the confidence of the causal direction [Mooij et al., 2016, Tagasovska et al., 2020]. The simple heuristic confidence [Mooij et al., 2016] is adopted here. For instance, for the proposed OCD, we define the confidence of $X \to Y$ to be $C_{X \to Y} = \text{BIC}(Y \to X | \boldsymbol{x}) - \text{BIC}(X \to Y | \boldsymbol{x})$.

**Results.** In Table 2, we summarize the ACC, AUC, and CPU times. For OCD and HCR, the average metrics over $L = 10, \ldots, 20$ as well as their standard errors are reported. The proposed OCD is highly competitive in all metrics. OCD has the second highest ACC and AUC, and is fast; it completes the analysis of 99 datasets in 36 seconds. Only IGCI, CAM, and LiNGAM are faster but they have worse ACC and AUC than OCD. SLOPE has slightly higher ACC and AUC than OCD. However, SLOPE is about 1 or 2 or-

ders of magnitude slower than OCD and relatively sensitive to small added noise (see the additional experiments that investigate the "Sensitivity to Small Added Noise" in the Supplementary Materials). Finally, the small standard errors of the performance metrics of OCD indicate its relative robustness with respect to the number $L$ of levels of discretization for the considered datasets and range.

Table 2: CEP data. Metrics of OCD and HCR are averaged over different values of $L = 10, \ldots, 20$ with standard errors given within the parentheses. Higher ACC and AUC are better.

|  | OCD | HCR | bQCD | CAM |
|---|---|---|---|---|
| ACC | 0.73 (0.01) | 0.44 (0.02) | 0.70 | 0.58 |
| AUC | 0.76 (0.00) | 0.56 (0.02) | 0.72 | 0.58 |
| CPU | 36s (1.7s) | 12m (2.2m) | 7m | 11s |
|  | IGCI | SLOPE | LiNGAM | RESIT |
| ACC | 0.66 | 0.76 | 0.42 | 0.53 |
| AUC | 0.51 | 0.84 | 0.59 | 0.56 |
| CPU | 1s | 24m | 3s | 12h |

## 6.4 SINGLE-CELL RNA-SEQUENCING DATA

We further validate the proposed OCD with a publicly available single-cell RNA-sequencing (scRNA-seq) dataset of $2,717$ murine embryonic stem cells [Klein et al., 2015]. We obtain a list of literature-curated pairs of transcription factor $(X)$ and its target $(Y)$ from the TRRUST database [Han et al., 2018], which provides biological ground truth of the casual relationships, namely $X \to Y$. We then extract the corresponding genes from the scRNA-seq dataset. Removing genes with more than 90% zeros (these genes have very low statistical variability), we retain 6701 pairs for causal validation, which still have 62% zeros. The zeros in scRNA-seq data are either (a) true biological zero counts or (b) small counts that are too low to detect. In either case, they can be regarded as "low expression". We compare OCD with the best performing methods in Section 6.3, bQCD and SLOPE, as well as the closest competitor HCR. We are not able to generate results (runtime errors) from CAM, LiNGAM, and RESIT possibly because of the large percentages of zeros. To apply OCD and HCR, we trichotomize the data at 0 and the median of the non-zero expression (i.e., "low", "average", and "high" expression). ACC and CPU time are reported in Table 3. OCD is the best and is the only method that is better than random guess (p-value = $10^{-75}$, binomial test with $H_0 : p = 0.5$ vs $H_a : p > 0.5$) for this dataset possibly because of its highly non-standard distribution due to zero-inflation. Therefore, although discretizing continuous or count data may lose information, it often improves the robustness by not having to impose a particular distributional assumption on the raw data.

Table 3: Single-cell RNA-seq data. Higher ACC is better.

|     | OCD | HCR | bQCD | SLOPE |
|-----|-----|-----|------|-------|
| ACC | 0.61 | 0.36 | 0.45 | 0.50 |
| CPU | 19m | 22m | 3.4h | 2h |

# 7   CONCLUSION

There are several limitations of the current work, which we plan to address in our future work. First, the current score-and-search algorithm outputs a point estimate of the causal graph with no uncertainty quantification and no global convergence guarantee. We plan to develop a fully Bayesian approach by assigning sparse priors (i.e., spike-and-slab priors on $\beta$'s) and carrying out posterior inference via the Markov chain Monte Carlo. Second, we have empirically assessed the identifiability of the proposed OCD for multivariate data and for bivariate data with unmeasured confounders. The identifiability theory for multivariate categorical data or bivariate categorical data with unmeasured confounders is in general lacking in the causal discovery literature. Third, we have not explicitly addressed the problem of choosing the number $L$ of categories in data discretization. We picked $L = 3$ for genomic data by convention and assessed its robustness up to $L = 10$. For non-genomic data, there is no obvious/universal choice of $L$. Instead of picking a specific $L$, we have tested the proposed OCD in a range of values. In the future, we plan to propose data-driven ways (e.g., via BIC) to objectively choose $L$.

**Acknowledgements**

Ni's research was partially supported by National Science Foundation (DMS-2112943 and DMS-1918851). Mallick's research was partially supported by TRIPODS National Science Foundation (CCF-1934904) and National Cancer Institute of the National Institutes of Health (R01CA194391).

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
