# OpenReview forum: "Ordinal Causal Discovery"
_auai.org/UAI/2022/Conference — UAI 2022 Poster_

### Official Review · Reviewer_VW8f · 2022-03-25

**Q2(1) Originality/Novelty:** 3
**Q2(2) Significance/Impact:** 2
**Q2(3) Correctness/Technical Quality:** 3
**Q2(6) Clarity Of Writing:** 4
**Q6 Overall Score:** 7
**Q8 Confidence In Your Score:** 4

**Q1 Summary And Contributions:**

The paper proposes a new method for causal discovery with purely ordinal data and applies this method to both synthetic and real data. The key contribution is the proposed method (because there are currently no other methods exploiting ordinality of data), and the identifiability result and the applications.

**Q2 Assessment Of The Paper:**

More detailed information regarding each of these aspects is given below:

**Q2(4) Quality Of Experiments (Optional):**

4: Excellent: The experimental evaluation is comprehensive and the results are compelling.

**Q2(5) Reproducibility:**

3: Good: Key resources (e.g., proofs, code, data) are available and key details (e.g., proofs, experimental setup) are sufficiently well-described for competent researchers to confidently reproduce the main results.

**Q3 Main Strengths:**

The proposed method constitutes a valuable addition to the field of causal discovery and the main strength definitely lies therein and in the accompanying identifiability result. Other than that, the paper is well written (save for a few grammatical errors here and there, but they are insignificant and rare) and it was a delight to read.

**Q4 Main Weakness:**

* It is not obvious to me that the bivariate identifiability result extends directly to dimensions of size $>2$ -- but perhaps I am missing something? If you think about usual identifiability results for ANMs, you usually have to jump through a few extra hoops to get them working in higher dimensions (e.g., the Peters et al. 2014 paper "Causal discovery with continuous additive noise models", Theorem 20 vs. Peters et al. 2014, Theorem 28).

**Q5 Detailed Comments To The Authors:**

First of all, congratulations on a very nice paper -- I thoroughly enjoyed reading it. Aside from my comment in Q4, I really only have one comment left:

* It would be nice to have a proof that minimizing the proposed score actually recovers the true graph, although I do not think this is in any way necessary for a submission like this. But it would be a nice addition in a future work.

**Q7 Justification For Your Score:**

The paper was very good, and I think it should be accepted, but I do not think it is likely to have a huge impact anywhere. Thus, the score.

**Q9 Complying With Reviewing Instructions:**

1: Yes.

---

### Official Review · Reviewer_rmeW · 2022-04-11

**Q2(1) Originality/Novelty:** 2
**Q2(2) Significance/Impact:** 3
**Q2(3) Correctness/Technical Quality:** 3
**Q2(6) Clarity Of Writing:** 4
**Q6 Overall Score:** 7
**Q8 Confidence In Your Score:** 4

**Q1 Summary And Contributions:**

The paper study the causal discovery problem with ordinal data.
In particular identifiability conditions are described and a method is proposed which can tackle both bivariate and multivariate causal discovery.
The identifiability is proven for the bivariate setting and the method is extended in multivariate setting using a greedy search.
The proposed methods are tested on synthetic data, on benchmark bivariate data and on real multivariate (discretized) data.

**Q2 Assessment Of The Paper:**

More detailed information regarding each of these aspects is given below:

**Q2(4) Quality Of Experiments (Optional):**

3: Good: The experimental evaluation is adequate, and the results convincingly support the main claims.

**Q2(5) Reproducibility:**

3: Good: Key resources (e.g., proofs, code, data) are available and key details (e.g., proofs, experimental setup) are sufficiently well-described for competent researchers to confidently reproduce the main results.

**Q3 Main Strengths:**

### Originality/Novelty/Significance/Impact
I think  (see Detailed comments) that the proof of identifiability is novel and surely interesting.
Causal discovery with ordinal data is an interesting subject and this paper advance the state of the art.

### Correctness/Technical Quality/Quality Of Experiments
The authors present a result of identifiability in the bivariate case with proof in the supplementary. The experiments are both on synthetic data and real world data.

### Reproducibility
Code for the proposed method is given in the supplementary as R package and the experiments are described in details.

### Clarity Of Writing
The paper is well written and is easy to follow.

**Q4 Main Weakness:**

### Originality/Novelty
One of my main concern of this paper is the relationship of the presented methods with the work of:
Ge Luo, Xiang, Giusi Moffa, and Jack Kuipers. "Learning Bayesian Networks from Ordinal Data." Journal of Machine Learning Research 22.266 (2021): 1-44.
(this was addressed by the authors in the rebuttal)

### Experiments

It is unclear ho the graphs in the simulation are generated (sec 6.1.1 and 6.1.2), It seems that for each number of nodes $p=5, 10, ..., 100$ only a single graph is randomly generated and then different data are sampled from BNs with the same graph and different coefficients $\beta_{jkl}, \alpha_j$.  It would be better to perform various repetition with different randomly generated graphs and report mean/median and standard deviations of the measures SHD and SID.
(this was addressed by the authors in the rebuttal)




**Q5 Detailed Comments To The Authors:**

### Previous work

On the relationship of the paper with previous work of Luo et al. 2021 (see Main Weakness for complete reference) I have the following comments/questions:

1. I think the model presented in this submission (with probit link) could be seen as one case of the model in Luo et al 2021 (Figure 1(c) there), is this true?
2. How the identifiability assumptions presented in this submission compare to identifiability in Luo et al. ?
3. Given the existence of this previous method, the proposed approach should be compared in the experiments (in the simulations at least) to the method of Luo et al. (code is available in R so it should not be a problem).

### Simulations and experiments

1. See Main weakness. By the sentence *The true BN is generated ran-
domly (Figure 2(a)) which has one v-structure* I understand that for each $p$ and dense or sparse scenario, the true graph is always the same. I think it is better to report results across various repetitions of the underlying graph random generation.

2. it would be useful to share the code of the simulations and the experiments, not only the code of the proposed method.

3. Luo et al applied their method to an ordinal psychological dataset (McNally et al . 2017), such dataset is a clear example of the type of data where ordinal observations are natural. A comparison of the methods in such data would be valuable.

### Other minor points

1. In section 3, definition 1 of distribution equivalence (DE) is given for bivariate OCD models. While in example 1 DE refers to two multinomial bivariate BN. Nevertheless example 1 is correct and I think everybody udnerstand what DE means in that case.




**Q7 Justification For Your Score:**

I think the paper is interesting, well written the method seems solid and the experiments are appropriate (see Weakness and Detailed comments for some details).
My main point of concern is the relationship of this work with a previous (but recent) work on causal discovery for ordinal data (see Weakness for the reference).

**Q9 Complying With Reviewing Instructions:**

1: Yes.

---

### Official Review · Reviewer_TEHP · 2022-04-12

**Q2(1) Originality/Novelty:** 2
**Q2(2) Significance/Impact:** 2
**Q2(3) Correctness/Technical Quality:** 3
**Q2(6) Clarity Of Writing:** 4
**Q6 Overall Score:** 7
**Q8 Confidence In Your Score:** 3

**Q1 Summary And Contributions:**

The paper aims to determine the direction of causal relationships for ordinal data. An ordinary causal discovery method is proposed to identify the causal structure. It compares the performance of the proposed method to alternative methods in both simulation and real data example.

**Q2 Assessment Of The Paper:**

More detailed information regarding each of these aspects is given below:

**Q2(4) Quality Of Experiments (Optional):**

3: Good: The experimental evaluation is adequate, and the results convincingly support the main claims.

**Q2(5) Reproducibility:**

3: Good: Key resources (e.g., proofs, code, data) are available and key details (e.g., proofs, experimental setup) are sufficiently well-described for competent researchers to confidently reproduce the main results.

**Q3 Main Strengths:**

1. The idea of making use of ordinary data in causal discovery has great potential in areas such as mental health, social survey, etc.
2. The quality of the experiments is good: many alternative methods are compared with the proposed methods; and three real data examples are provided.
3. The structure is clear and easy to follow.

**Q4 Main Weakness:**

The paper considers the case where all variables are ordinal and suggests that continuous variable can be converted to ordinal variable. If $X$ is continuous and $Y$ is ordinal, does the identification still hold? It seems that $X$ and $Y$ are no longer symmetric and the assumptions in ordinal regression model may be violated?

**Q5 Detailed Comments To The Authors:**

1. Is Definition 1 defining distribution equivalence only for ordinal data? Is the definition compatible for Example 1?
2. In Algorithm 1, how is the initial graph G determined?

**Q7 Justification For Your Score:**

The idea to use ordinal variables to determine the direction fo causal relationships is inspiring. The technical details seems to be good. The experiments are adequate.

**Q9 Complying With Reviewing Instructions:**

1: Yes.

---

### Official Review · Reviewer_2Df3 · 2022-04-12

**Q2(1) Originality/Novelty:** 3
**Q2(2) Significance/Impact:** 2
**Q2(3) Correctness/Technical Quality:** 3
**Q2(6) Clarity Of Writing:** 4
**Q6 Overall Score:** 7
**Q8 Confidence In Your Score:** 3

**Q1 Summary And Contributions:**

This paper proposes a method to discover for causal relationships between ordinal categorical variables. The method proposed in this study is applicable to multivariate data as well as bivariate data. Existing studies have proposed methods based on conditional independence, but those methods cannot distinguish Markov equivalence graphs. This paper proposes a method to solve such a problem.

**Q2 Assessment Of The Paper:**

More detailed information regarding each of these aspects is given below:

**Q2(4) Quality Of Experiments (Optional):**

3: Good: The experimental evaluation is adequate, and the results convincingly support the main claims.

**Q2(5) Reproducibility:**

3: Good: Key resources (e.g., proofs, code, data) are available and key details (e.g., proofs, experimental setup) are sufficiently well-described for competent researchers to confidently reproduce the main results.

**Q3 Main Strengths:**

Many causal analyses involve ordinal categorical variables in the data. Therefore, the method proposed in this study has an important impact in causal search. The effectiveness of the method is demonstrated through various experiments in this paper. The method can also be applied for the analysis of causal relationships between ordinal categorical variables and continuous variables if the continuous variables are ordinalized by their values. Although the paper assumes the absence of unobserved common causes, experiments show that the proposed method is robust to unobserved common causes. Experiments also show that the proposed method is effective for continuous variables that are replaced by discrete variables according to their values.

**Q4 Main Weakness:**

The identifiability of causality between two variables is shown in the theorem, while for multi variables, a theorem is needed to show that only one causal graph is generated from data having multi variables.

Experiments show that the proposed method is robust against unobserved common causes, but theoretical reasons should also be provided.

As the authors state, methods using hidden compact representation have so far been limited to causal relationships between two variables. However, Jie Qiao et al. 2021 "Learning causal structures using hidden compact representation" proposes a method for estimating causal relationships among multivariate discrete variables. This method should also be discussed.

**Q5 Detailed Comments To The Authors:**

As for the identifiability theorem for multivariable graphs, I do not consider it essential, but if it can be added, please add it.

Why is the proposed method robust even when there is an unobserved common cause?

Please also discuss Jie Qiao et al. 2021 "Learning causal structures using hidden compact representations".

**Q7 Justification For Your Score:**

This paper has an important impact because causal discovery among ordinal categorical variables is important while methods for them had never been proposed. The effectiveness of the method proposed in this paper has been amply demonstrated by experimentation.

**Q9 Complying With Reviewing Instructions:**

1: Yes.

---

### Official Review · Reviewer_Vkmo · 2022-04-14

**Q2(1) Originality/Novelty:** 3
**Q2(2) Significance/Impact:** 3
**Q2(3) Correctness/Technical Quality:** 3
**Q2(6) Clarity Of Writing:** 3
**Q6 Overall Score:** 6
**Q8 Confidence In Your Score:** 4

**Q1 Summary And Contributions:**

The authors propose a simple method for dealing with the important problem of causal discovery in the setting of ordinal categorical variables. They provide extensive experimental justification for OCD and their method development is sound and easy to follow.

**Q2 Assessment Of The Paper:**

More detailed information regarding each of these aspects is given below:

**Q2(4) Quality Of Experiments (Optional):**

4: Excellent: The experimental evaluation is comprehensive and the results are compelling.

**Q2(5) Reproducibility:**

3: Good: Key resources (e.g., proofs, code, data) are available and key details (e.g., proofs, experimental setup) are sufficiently well-described for competent researchers to confidently reproduce the main results.

**Q3 Main Strengths:**

- Interesting problem
- Good proposed solution
- Good empirical evaluation
- Well written, paced and formulated paper

**Q4 Main Weakness:**

- See main review

**Q5 Detailed Comments To The Authors:**

# Introduction

- An example which I like to use is comparing a sample of red, yellow and green. It could come from a bag of Smarties or from a traffic light. The former is categorical but the latter is ordinal.
- The using quotation marks, the latex engine requires you to write it properly as `` '' it will not parse properly if you write " "
- If you were to relax the sufficiency assumption, what would OCD do? Or rather why do you assume causal sufficiency and more what would have to do to adapt OCD to handle unobserved confounders?

# BIVARIATE ORDINAL CAUSAL DISCOVERY

- Why are you using different density notation in eq 1 and 2?

# Identifiability

- Figure 1 is ... rather ugly. There are some excellent packages for parsing python/R/excel data into latex format. I suggest you use one of those to make the exposition of your work more appealing to the eye.
- Please explain "for almost all" in theorem 1. What are the cases that are not included in that sentence? Is this the binary case which you talk about later in this section?

# EXTENSION TO MULTIVARIATE ORDINAL CAUSAL DISCOVERY

- Avoid using $p$ for anything but probabilistic quantities, use anything else as to avoid confusion

# CAUSAL GRAPH STRUCTURE LEARNING

-  Algorithm 1 can find the local optima, but not the global optima? Is that correct? Is that useful? How could you extend it to find the global optima in the search-space?

# EXPERIMENTS

-  Figures 2 and 3 are great, very nicely summarises OCD and its performance compared to other methods (though again, please use anything but p for the number of nodes)
- It would help if you bolded relevant (best) results in your tables.
- Tell us, in the legend of the tables, what is good (is low good or is a high metric good), so that we don't have to dive into the text where you explain this once. This way you make life easier for your reader.

**Q7 Justification For Your Score:**

I like this paper. It is interesting and useful. The score is low because it is on the empirical side and UAI tends to be more methods/theory focused. That being said, I am happy to increase my score if my questions are adequately addressed as this is an interesting problem and a good method has been proposed for tackling it.

**Q9 Complying With Reviewing Instructions:**

1: Yes.

---

### Decision · Program_Chairs · 2022-05-15

**Decision:**

Accept (Poster)

**Comment:**

Meta Review: This paper considers causal direction identification and structure learning with ordinal data. They formulate a probabilistic model of one ordinal variable conditional on another (generalizing e.g. probit and inverse logit regressions) and show that "typically" (wrt Lebesgue measure), data consistent with the X->Y direction of such a model cannot be described by the Y->X direction. This opens up possibilities for causal identification that would not be missed if the data were simply treated as categorical.

Pros:

Reviewers felt this was an interesting, novel, and relevant approach. They argued that categorical data is indeed common in many applications and the general view was that this line of work is therefore promising. They asked for clarifications on the paper and the authors responded in detail and constructively, identifying also relevant areas for future work.

Cons:

Reviewers pointed out it is not obvious how this work generalizes to multiple dimensions or latent confounding. The authors agreed that these extensions are not obvious and will require further work. Nevertheless reviewers felt that the contributions of this paper were substantial enough on their own.